# Differential Property Prediction: A Machine Learning Approach to Experimental Design in Advanced Manufacturing

**Loc Truong[1], WoongJo Choi[1], Colby Wight[1], Lizzy Coda[1], Tegan Emerson[1,2], Keerti Kappagantula[1], Henry Kvinge[1,3]**

[1]Pacific Northwest National Laboratory
[2]Department of Mathematics, Colorado State University
[3]Department of Mathematics, University of Washington
{first}.{last}@pnnl.gov

## Abstract

Advanced manufacturing techniques have enabled the production of materials with state-of-the-art properties. In many cases however, the development of physics-based models of these techniques lags behind their use in the lab. This means that designing and running experiments proceeds largely via trial and error. This is sub-optimal since experiments are cost-, time-, and labor-intensive. In this work we propose a machine learning framework, differential property classification (DPC), which enables an experimenter to leverage machine learning's unparalleled pattern matching capability to pursue data-driven experimental design. DPC takes two possible experiment parameter sets and outputs a prediction of which will produce a material with a more desirable property specified by the operator. We demonstrate the success of DPC on AA7075 tube manufacturing process and mechanical property data using shear assisted processing and extrusion (ShAPE), a solid phase processing technology. We show that by focusing on the experimenter's need to choose between multiple candidate experimental parameters, we can reframe the challenging regression task of predicting material properties from processing parameters, into a classification task on which machine learning models can achieve good performance.

## Introduction

Despite impressive progress in tasks ranging from object recognition, to speech-to-text, to games such as Go (Silver et al. 2017), there are many scientific domains where machine learning (ML) is just beginning to have a significant impact. A striking example of the potential ML has for transforming the sciences was recently demonstrated with the success of AlphaFold for the problem of predicting protein folding (AlQuraishi 2019). While advanced manufacturing also has many challenges that would benefit from the strong pattern matching capabilities of machine learning systems, the intersection of these two fields is still in its infancy (Arinez et al. 2020). In this work, we propose a machine learning-based framework to aid in experimental design in advanced manufacturing.

Because of the physical regimes in which they process materials, advanced manufacturing techniques frequently lack physics-based models that can be used to choose favorable experiment processing parameters. This is a significant limitation because without such models as a guide, trial and error methods have to be used to manufacture samples with desired performance metrics which results in less efficient research and development. Thus, there is a significant need to develop predictive methods that can help guide the experimenter toward processing parameters that will help them optimize a specific property.

We call our framework differential property classification (DPC). A DPC model is designed to distinguish between two sets of process parameters, identifying which (if any) will result in a material with a larger property value. For example, the process parameters for some manufacturing process may be the temperature to which a material is heated or the pressure that is exerted on it during manufacturing. A property of the resulting material may be ultimate tensile strength (UTS). In such an example, DPC would help the experimenter identify those temperature and pressure values that will result in a material with high (or low) UTS. Of course, a DPC model is specific to a particular manufacturing technique, a particular material system, and a particular property $Y$. It takes as input two sets of manufacturing processing parameters $A$ and $B$ and as output provides a prediction of whether (1) processing parameters $A$ will yield a material with higher property $Y$ than processing parameters $B$, (2) processing parameters $B$ will yield a material with higher property $Y$ than processing parameters $A$, or (3) the processing parameters $A$ and $B$ will yield a material with approximately the same value for property $Y$ (see Figure 1). The idea is that when deciding between a range of possible experiments to run, the experimenter can use DPC to select the set of processing parameters that optimizes for the desired property.

The motivation for translating what might otherwise be a standard regression problem ("what is the value of property $Y$ for sample produced using process parameters $A$?") into a 3-way classification problem, comes from two observations. The first observation is that there is frequently only a limited amount of data associated with advanced manufacturing processes. Classification problems often require less data to achieve an acceptable level of accuracy than regression problems do. If one can solve a problem in an easier classification setting as opposed to a more challenging re-

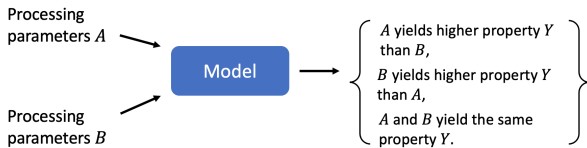

Figure 1: A schematic of the DPC model. DPC helps an experimenter choose between possible processing parameters for a manufacturing process.

gression setting, then one should choose the former.

The second related observation is that in designing experiments in the materials and manufacturing domain, identifying relative performance of materials produced from a range of candidate process parameters is more valuable than the exact material properties that will result from each. This is especially true in the case where the former can be done with strong accuracy while the latter cannot due to the size of the data set. Since domain scientist trust is an essential component of building a machine learning tool that will be used, it is critical that we solve the problem that needs to be solved rather than over-promising and under-delivering and thus losing scientist trust. In this case, this means building a DPC model that achieves high accuracy instead of a regression model whose performance is less satisfactory.

We demonstrate the effectiveness of DPC on a real-world advanced manufacturing dataset consisting of the process conditions/mechanical properties measurements from 20 experiments of AA7075 tubes synthesis using Shear Assisted Processing and Extrusion (ShAPE) (Whalen et al. 2021b,a) to aluminum 7075. We explore a range of different model types and training regimes, highlighting those that result in the best performance. We also analyze our model with respect to variable amounts of training data, showing that DPC models are relatively robust even when only small amounts of data are available. This is an important property since the purpose of DPC is to guide experimentation and thus our assumption should always be that DPC will be used in situations where little data currently exists.

## Related Work

The ability to predict material properties from manufacturing conditions is a critical capability in advanced manufacturing. Aside from improving the quality of a final product, it can also accelerate the research and development cycle by enabling experimenters to efficiently find processing parameters that produce a desired material property.

Recent examples of this include (Li et al. 2019) where a range of techniques were used to predict the surface hardness of printed parts based on processing parameters in a material extrusion process. In a similar direction, (Lao et al. 2020) developed models which predicted extruded surface quality based on processing parameters in 3D printing of concrete. (Mohamed, Masood, and Bhowmik 2017) used a neural network to optimize for viscoelastic responses in a Fused Deposition Modelling (FDM) 3D Printing process. In

(Jiang et al. 2020), on the other hand, a framework was developed to predict properties from process parameters and vise versa for a customized ankle bracelet with tunable mechanical performance with stiffness. These and other works use a range of model types from decision trees to neural networks to predict properties.

To our knowledge, our work is the first to propose an alternate classification framework for process parameter/property prediction which is better adapted to low-data regimes while still serving the needs of a material/manufacturing scientist.

## The DPC Framework and Model

The DPC framework involves translating what would naively seem to be a regression problem, into a classification problem on pairs of process parameters. Suppose that $X$ is the set of all possible process parameters for a given manufacturing process, $Y = \mathbb{R}$ is the set of all possible material property values for a given property, $D_t = \{(x_i^t, y_i^t)\}_{i=1}^{k_1}$ is a process parameter/property regression training set, and $D_e = \{(x_i^e, y_i^e)\}_{i=1}^{k_2}$ the corresponding regression test set. We choose some $t \in \mathbb{R}$ which will be the threshold we use to identify whether two property values $y_1$ and $y_2$ are "different". The DPC test set associated with this task is:

$$\widetilde{D}_e = \{(x_{i_1}^e, x_{i_2}^e, z_{i_1,i_2}) \mid 1 \leq i_1, i_2 \leq k_2, z_{i_1,i_2} \in Z\} \quad (1)$$

where $Z = \{0, 1, 2\}$ are the classes and

$$z_{i_1,i_2} = \begin{cases} 1 & \text{if } y_{i_1}^e - y_{i_2}^e > t, \\ 2 & \text{if } y_{i_2}^e - y_{i_1}^e > t, \\ 0 & \text{if } |y_{i_1}^e - y_{i_2}^e| < t. \end{cases} \quad (2)$$

The latter case, where the absolute difference between $y_{i_1}$ and $y_{i_2}$ is less that $t$, can be interpreted as describing when $y_{i_1}$ and $y_{i_2}$ are sufficiently close so as to be treated as the "same". This could be because property measurements are noisy or because two measurements might as well be the same from a practical standpoint. For example, if two samples have a max load of 1739.4kg and 1739.9kg respectively, we might not consider them different from the standpoint of this material property. We can build a validation or training set in a manner analogous to that described above.

Once a test set, $\widetilde{D}_e$, has been constructed, we choose a machine learning model capable of doing 3-way classification. The DPC framework is agnostic to the particular model architecture and different model types may be preferable depending on the nature of the data. Since we were working with relatively low-dimensional data our experiments in this paper used eXtreme Gradient Boosting (XGBoost) (Chen and Guestrin 2016), a tree-based boosting algorithm, and a simple feed-forward neural network. Training can be done by training a backbone model to do regression and then inserting it into the DPC framework, by training a DPC model to do classification directly, or some combination of the two.

The choice of $t$ should largely be driven by the application. If $t$ is too small, pairs of process parameters that do not actually result in meaningfully different material properties will be labelled as if they do. If $t$ is too large, legitimately

different property values may be grouped as if they were the same. Furthermore, as $t$ changes the class balances will shift. When $t = 0$, there are no elements from class '0' other than identical pairs. On the other hand, when $t$ is large class '0' dominates. In the experiments below we frequently chose $t$ to be some fraction of the standard deviation of property values, for example $1\%$ of standard deviation.

## Experiments

We trained and evaluated our DPC models on data collected from AA7075 tube mechanical properties and corresponding processing conditions. The tubes were manufactured using ShAPE, a solid phase processing technique (Whalen et al. 2021a,b). During ShAPE, a rotating die impinges on a stationary billet housed in an extrusion container with a coaxial mandrel. Due to the shear forces applied on the billet as well as the friction at the tool/billet interface, the temperature increases, and the billet material is plasticized. As the tool impinges into the plasticized material at a predetermined feed rate, the billet material emerges from a hole in the extrusion die to form the tube extrudate. AA7075 tubes were manufactured using ShAPE at different tool feed rates and rotation rates using homogenized and unhomogenized AA7075 castings. The tubes were subsequently tempered to T5 and T6 conditions and then their mechanical properties, namely ultimate tensile strength (UTS), yield strength (YS), % elongation were tested.

### The Training and Test Set

The dataset that we used for training and testing is comprised of 20 distinct ShAPE experiments. Each experiment resulted in a single extruded aluminum 7075 tube. Some process parameters such as mechanical power, extrusion torque, tool position with respect to billet, extrusion force, and extrusion temperature were measured continuously (every .01 seconds) over the course of the ShAPE experiment resulting in time series. Others such as heat treatment time are available as discrete data points.

Material properties were measured for samples obtained from (on average) 10 locations along the length of an extruded tube. Since there are in general many more process parameter measurements than material property measurements, the size of our dataset is limited by the number of material properties that were measured.

We split our dataset at the level of individual experiment into $75\%$ (15 experiments) for the training set $D_t$ and $25\%$ (5 experiments) for the test set $D_e$. Note that since process parameters and properties measured across the tube produced in a single experiment are frequently similar, if we were to mix measurements from a single experiment between training and test sets we would risk the models memorizing characteristics particular to each experiment. We constructed a corresponding classification test set $\widetilde{D}_e$ following description (1). This involved generating all possible pairs of process parameter/property data points from $D_e$ resulting in 1600 pairs in $\widetilde{D}_e$. We also generated the new labels from $Z$. For one of our models we generated a classification set $\widetilde{D}_t$ from $D_t$ for training. For all experiments in the paper we used a threshold $t$ equal to $1\%$ of the standard deviation of measurements for the particular property value.

## Models and Training

The backbone models we used in our experiments differed along two dimensions: model architecture and model type. By model architecture we mean the base learning algorithm underlying the DPC model. We explored two of these. The first is a multilayer perceptron (MLP), i.e., a vanilla feedforward neural network with fully-connected layers and nonlinearities. All of our MLPs were trained using the Adam optimizer with a learning rate of $0.009$. While we experimented with other network architectures, the primary one that we used across several experiments has 3 layers including a hidden layer of dimension 35. We used ReLU nonlinearities in all cases. The second model architecture we tested was an XGBoost decision tree model that was trained with a max depth of 6 and 1000 estimators at a $0.1$ learning rate. We used Pytorch (Paszke et al. 2019) to implement the MLP.

We explored three different backbone model types. The first, which we call a *direct regression model* takes a regression model $f : X \rightarrow Y$ that has been trained on $D_t$ and use it to predict values from $Z$. That is, for input pair $(x_1, x_2, z) \in \widetilde{D}_e$, we calculate $f(x_1)$ and $f(x_2)$ and predict $z$ based on their values in accordance with (2). The second backbone model type we explored, which we call the *difference regression model*, is trained so that given input $(x_1, y_1) \in D_t$ and $(x_2, y_2) \in D_t$, model $f : X \times X \rightarrow Y$ predicts the difference $y_1 - y_2$. This difference prediction can again be used to predict a value from $Z$ via (2). The final model type that we explored was a *direct classification model*. Models of this type take concatenated pairs of process parameters from $(x_1, y_1)$ and $(x_2, y_2)$, and predict the corresponding label from $Z$ directly.

Note that all of these model types use different forms of the training set. Direct regression models are trained on $D_t$. On the other hand, difference regression models are trained on a derivation of $D_t$ which is constructed from pairs of process parameters. The target value in this case is material property differences. The direct classification models are trained on $\widetilde{D}_t$, which is constructed from $D_t$ analogously to what is outlined in (1) and (2). Direct regression and difference regression models are trained with respect to mean squared error (MSE), while direct classification models are trained with cross entropy.

## Results and Discussion

We begin by evaluating the performance of the two different architectures underlying our DPC models (MLPs and XGBoost models). Table 1 contains the accuracies for a direct regression backbone version of each model on the test set $\widetilde{D}_e$. We include $95\%$ confidence intervals for the MLP which had more variable performance based on the random weight initialization. These intervals were calculated over 5 different random initializations. We see that the XGBoost model achieves consistently better performance than the MLP for each of the three material properties that we evaluated. Particularly striking is the comparison between the

Table 1: The accuracy of both DPC models (MLP and the XGBoost model) on the test sets for different material properties. We include $95\%$ confidence bounds which are calculated over 5 random weight initializations of the MLP.

|  | MLP | XGBoost |
|---|---|---|
| Max Load | $77.00 \pm 3.0$ | **87.81** |
| UTS | $88.00 \pm 1.0$ | **89.00** |
| Yield Strength | $79.00 \pm 1.0$ | **82.94** |

Table 2: DPC accuracy values for different backbone model types: direct regression, difference regression, and direct classification. The first two models were trained for a regression task, while the last was only trained for DPC prediction. All backbone models use XGBoost, our best performing architecture (see Table 1).

|  | Max load | UTS | Yield Strength |
|---|---|---|---|
| Direct reg. | 86.12 | **90.00** | 79.00 |
| Difference reg. | 84.56 | 86.50 | 77.00 |
| Classification pred. | **87.81** | 89.00 | **82.00** |

XGBoost and MLP models performance predicting which process parameters would result in a material with greater max load. In this case the XGBoost model achieves accuracy almost $10\%$ better than the MLP. We hypothesize that the XGBoost model's superior performance arises from it being a simpler model that is less likely to overfit to the small training sets that were used.

We next compared the different backbone model types (direct regression, difference regression, and direct classification) that were described in the Models and Training section. Results from our experiments are shown in Table 2. We see that overall, direct regression and direct classification appear to perform similarly with both methods delivering comparable accuracy on the three different properties. On the other hand, difference regression consistently underperformed relative to the other two methods.

We believe that there are two factors in play here. On the one hand, models trained on the regression task are exposed to additional information that models trained only on classification are not. For example, a regression model learns patterns relating training process parameters $x$ to its absolute associated material property $y$, whereas the classification model only learns a relative comparison and does not see the property magnitudes themselves. On the other hand, the direct classification model has been optimized for the final task that it will be evaluated on, whereas the direct regression model is optimized for a different (though related) task.

We suspect that a model that is more robust than either the direct regression or direct classification types could be developed by designing a loss function that includes the raw material property values while still directly optimizing for accuracy in the DPC task. This was our goal with the difference regression model, but experiments showed that this

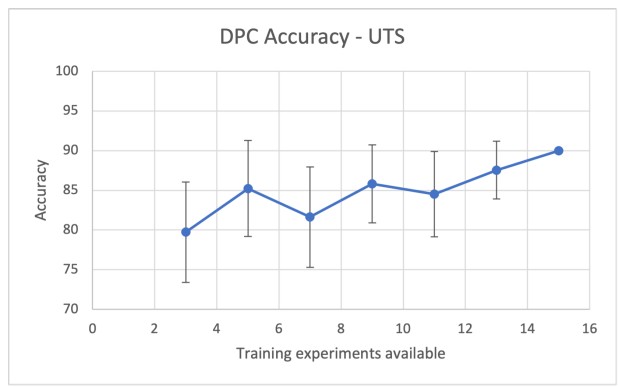

Figure 2: A comparison of DPC accuracy (for an XGBoost direct regression backbone model) on the test set based on the number of experiments in the training set. Recall that there are 15 experiments, each experiment provides around 10 process parameter/property pairs for the training set. We created error bars by randomly sampling and then training on 5 different size $k$ subsets for each of $k = 3, 5, \ldots, 15$.

approach did not fully harness the strengths of both versions.

Finally, given that DPC was developed to be able to work in low-data environments, we wanted to explore how DPC accuracy changes as the number of experiments available for training changes. In Figure 2 we plot the accuracy of a DPC model that uses an XGBoost direct regression backbone model on the fixed test set as a function of the number of experiments in the training set. Recall that each experiment contributes (roughly) 10 process parameter/property pairs to the training set. We see that even in the ultra-low data regime of 5 experiments, the model still achieves reasonable accuracy of $80\%$. The model's performance continues to improve, reaching $90\%$ at 15 experiments. The amount of variability also decreases significantly as can be seen by the error bars that represent multiple runs over random subsets of the training set. We note that one of the benefits to ML-driven experiment planning is that the model quickly becomes better at guiding experiments at more experiments are performed, resulting in a convenient positive feedback loop.

## Conclusion

In this work we presented a new framework, differential property classification (DPC), to aid in experiment planning in advanced manufacturing. DPC is designed to handle one of the persistent challenges of working with machine learning in the field of advanced manufacturing: limited amounts of data. Through our experiments using real ShAPE data, we showed that DPC can yield helpful predictions even when very few experiments have already been run. We believe that this represents another step toward the larger goal of leveraging data-driven methods to improve efficiency of the advanced manufacturing research and development cycle.

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

## Acknowledgments

KSK thanks Scott Whalen, Md. Reza-E-Rabby, Tianhao Wang and Timothy Roosendaal for their insights into AA7075 manufacturing and property determination. KSK is grateful for the discussions on advanced manufacturing with Cindy Powell and Glenn Grant.

This research was supported by the Mathematics for Artificial Reasoning in Science (MARS) initiative via the Laboratory Directed Research and Development (LDRD) investments at Pacific Northwest National Laboratory (PNNL). PNNL is a multi-program national laboratory operated for the U.S. Department of Energy (DOE) by Battelle Memorial Institute under Contract No. DE-AC05-76RL0-1830.
