# OpenReview forum: "Differential Property Prediction: A Machine Learning Approach to Experimental Design in Advanced Manufacturing"
_AAAI.org/2022/Workshop/ADAM — AAAI 2022 Workshop ADAM_

### Official Review · Reviewer_MKyT · 2021-12-01
**Not fully convinced of the motivation of the approach, but well-written paper.**

**Rating:** 6
**Confidence:** 3

**Review:**

This paper provides the differential property classification framework (DPC), which converts the regression into a classification problem. The author proposes labeling three different classes based on the difference of two input processing parameters and verifying their method on AA7075 tube properties.

The main concern of this paper is their justification of converting into a DPC model rather than a naive regression problem. The authors claim that "classification problems often require fewer data to achieve an acceptable level of accuracy than regression problems do"; however, this statement has no guarantee by just considering the dataset following basic linear models. Furthermore, the author will need to provide more systematic approach to support their argument since this is the primary motivation for leveraging the DPC model.

Furthermore, in the section "the DPC Framework and Model," the authors give an example "if two samples have a max load of 1739.4kg and 1739.9kg respectively, we might not consider them different from the standpoint of this material property" to support to use DPC. However, if the choice of $t$ needs to be large enough as a hyperparameter, I do not see the apparent motivation of converting the regression problem to a classification problem again.

Still, I see the paper's topic fits into the workshop, and I give the rating marginally above the acceptance threshold.

---

### Official Review · Reviewer_ARNA · 2021-12-03
**Differential property prediction**

**Rating:** 5
**Confidence:** 3

**Review:**

Authors use XGBoost and MLP to predict property differences for different material pairs. While the paper is for most part well written, I am not convinced of the motivation, novelty, and experimental setting. The current problem can be handled using a learning to rank model, for which one can use simple (such as logistic regression) to complex (DNN) model) - an area that has been widely studied. Second, it is not clear how many samples were used for training/testing and what is input dimension. The number of training/test samples are 15/5 as reported. Then the paper states "Material properties were measured for samples obtained from (on average) 10 locations along the length of an ex- truded tube." Does that mean that for each experiment there were 10 samples generated? Even then,  it is not clear how authors ensure that the XGBoost or a 3 layer MLP does not overfit and how do they handle curse of dimensionality.